**www.cambridge.org/qrd**

# Thermodynamic control of gene regulation

James W. Wells and Tigran V. Chalikian 🄳

Department of Pharmaceutical Sciences, Leslie Dan Faculty of Pharmacy, University of Toronto, Toronto, ON, Canada

## Perspective

conformational control of transcription; distribution of conformational states; G-quadruplex; thermodynamics; *i*-motif

**Corresponding author:**
Tigran V. Chalikian;
Email: t.chalikian@utoronto.ca

## Abstract

G-quadruplexes and *i*-motifs are non-canonical secondary structures of DNA that act as conformational switches in controlling genomic events. Within the genome, G- and C-rich sequences with the potential to fold into G-quadruplexes and *i*-motifs are overrepresented in important regulatory domains, including, but not limited to, the promoter regions of oncogenes. We previously have shown that some promoter sequences can adopt coexisting duplex, G-quadruplex, *i*-motif, and coiled conformations; moreover, their distribution can be modelled as a dynamic equilibrium in which the fractional population of each conformation is determined by the sequence and local conditions. On that basis, we proposed a hypothesis in which the level of expression of a gene with G- and C-rich sequences in the promoter is regulated thermodynamically by fine-tuning the duplex-to-G-quadruplex ratio, with the G-quadruplex modulating RNA polymerase activity. Any deviation from the evolutionarily tuned, gene-specific distribution of conformers, such as might result from mutations in the promoter or a change in cellular conditions, may lead to under- or overexpression of the gene and pathological consequences. We now expand on this hypothesis in the context of supporting evidence from molecular and cellular studies and from biophysico-chemical investigations of oligomeric DNA. Thermodynamic control of transcription implies that G-quadruplex and *i*-motif structures in the genome form as thermodynamically stable conformers in competition with the duplex conformation. That is in addition to their recognized formation as kinetically trapped, metastable states within domains of single-stranded DNA, such as a transcription bubble or R-loop, that are opened in a prior cellular event.

## Introduction

An understanding of transcriptional regulation is arguably one of the most important challenges in molecular biology (Lee and Young, 2000). The initiation of transcription is controlled by promoters, which serve as binding sites for RNA polymerases, transcription factors, and other proteins of the transcriptional machinery. Although the sequence-specificity of all DNA-binding proteins, including transcription factors, tends to be very high, some transcription factors may recognize their target sites on promoters by combining sequence selectivity with structural recognition. This possibility came to light with the discovery of non-canonical G-quadruplex and *i*-motif structures in the promoter regions of many genes, including—most importantly—oncogenes (Brooks *et al.*, 2010; Balasubramanian *et al.*, 2011; Hansel-Hertsch *et al.*, 2017).

In addition to the B-DNA duplex, genomic DNA may adopt various non-canonical conformations such as Z-DNA, triplex DNA, cruciform DNA, G-quadruplexes, and *i*-motifs (Duckett *et al.*, 1995; Frank-Kamenetskii and Mirkin, 1995; Plum *et al.*, 1995; Lane *et al.*, 2008; Choi and Majima, 2011; Hansel-Hertsch *et al.*, 2017; Spiegel *et al.*, 2020; Sugimoto *et al.*, 2021; Tateishi-Karimata and Sugimoto, 2021). Guanine (G)- and cytosine (C)-rich DNA strands in particular may form G-quadruplexes and *i*-motifs, respectively, which are four-stranded secondary structures whose basic unit is a G-tetrad (G-quadruplex) or a hemi-protonated pair of cytosines (*i*-motif) (Lane *et al.*, 2008; Balasubramanian *et al.*, 2011; Bochman *et al.*, 2012; Benabou *et al.*, 2014; Day *et al.*, 2014; Obara *et al.*, 2024). The structure and energetics of DNA states, both canonical and non-canonical, and the enthalpic and entropic interactions that govern the stability of such states, have been the subject of many reviews (*e.g.,* Frank-Kamenetskii and Mirkin, 1995; Plum *et al.*, 1995; Lane *et al.*, 2008; Khutsishvili *et al.*, 2009; Benabou *et al.*, 2014; Day *et al.*, 2014; Nakano *et al.*, 2014; Privalov and Crane-Robinson, 2018; Vologodskii and Frank-Kamenetskii, 2018).

The human genome contains hundreds of thousands of G- and C-rich sequences with the potential to fold into a G-quadruplex or an *i*-motif (Spiegel *et al.*, 2020; Varshney *et al.*, 2020; Tateishi-Karimata and Sugimoto, 2021). The distribution of those structures in the genome is not random; rather, they are overrepresented in loci of critical importance, including, but not limited to, the promoters of oncogenes (Balasubramanian *et al.*, 2011; Chen *et al.*, 2022; Romano *et al.*, 2023; Zanin *et al.*, 2023). Tumor-related genes, such as c-MYC, hTERT, c-*kit*, KRAS, Bcl-2, and VEGF, have been identified as genes in which a G-quadruplex is formed and is involved in transcriptional regulation (Waller *et al.*, 2009; Balasubramanian *et al.*, 2011; Alessandrini *et al.*,

2021; Kosiol *et al.*, 2021; Robinson *et al.*, 2021; Chen *et al.*, 2022; Romano *et al.*, 2023; Zanin *et al.*, 2023).

There is a widespread effort to understand the transcriptional role of G-quadruplexes and *i*-motifs. Accumulating evidence suggests that those four-stranded non-canonical structures act as stimulators or inhibitors of transcription, with the balance between the two effects being fine-tuned for each gene and cell cycle (Kendrick *et al.*, 2014; Kim, 2019; King *et al.*, 2020; Lago *et al.*, 2021; Robinson *et al.*, 2021). In one striking example, a G-quadruplex, but not its duplex counterpart, serves as the primary recognition site for key transcription factors and chromatin proteins that bind to the c-MYC promoter (Esain-Garcia *et al.*, 2024). Loss of the G-quadruplex leads to suppression of c-MYC transcription, which can be restored by replacing the endogenous G-quadruplex with a G-quadruplex from the KRAS oncogene (Esain-Garcia *et al.*, 2024). Thus, controlled formation or resolution of a G-quadruplex in a promoter is a mechanism of transcriptional control (Robinson *et al.*, 2021). It has been suggested that G-quadruplexes and *i*-motifs are both involved in the regulation of transcription, albeit through different mechanisms (Zanin *et al.*, 2023).

The functioning of tetraplex DNAs in the genome relates to the nature of their formation, which remains an open question. In one scenario, a G-quadruplex or *i*-motif forms as a thermodynamically stable state that competes with the duplex conformation; in another, the tetraplex may occur as a kinetically stabilized metastable state within a pre-dissociated single-stranded stretch of genomic DNA. The lack of an answer to this question hampers our understanding of the conformational control of transcription and the tetraplex-dependent modulation of RNA polymerase activity.

Studies in our laboratory have suggested that the B-DNA and G-quadruplex conformations in a promoter may coexist in a site-specific dynamic equilibrium, in contrast to the prevailing view that they occur exclusively as one form or the other (Liu *et al.*, 2020, 2022; Garabet *et al.*, 2025). Those results gave rise to a hypothesis in which the level of gene expression is regulated in an essentially thermodynamic manner through a fine-tuning of the ratio of duplex to G-quadruplex, with the G-quadruplex acting as a conformational on- and off-switch modulating the activity of RNA polymerase. The fine-tuning is achieved by the evolution-selected promoter sequence and the "native" intracellular conditions, including the pH and the concentrations of $K^+$ and $Na^+$ ions. Any deviation from the native distribution of conformations, which could result from a point mutation or a disease-induced change in cellular conditions, may be accompanied by under- or overexpression of the gene.

Such a thermodynamic hypothesis of transcriptional control explains, is consistent with, and is supported indirectly by several observations. (i) Studies on oligomeric constructs *in vitro* suggest that G-quadruplexes and *i*-motifs can coexist with the duplex in a thermodynamic equilibrium, with the fractional populations dependent upon the nucleotide sequence and local conditions (Chalikian *et al.*, 2020; Liu *et al.*, 2020, 2022; Garabet *et al.*, 2025). Extending this observation to chromatin, G- and C-rich domains in the genome similarly may fluctuate between the duplex and tetraplex conformations in a site-specific dynamic equilibrium. (ii) G-quadruplex and *i*-motif structures form in promoters, both *in vitro* and naturally in cellular DNA (Balasubramanian *et al.*, 2011; Tateishi-Karimata and Sugimoto, 2020; Lago *et al.*, 2021; Zanin *et al.*, 2023). (iii) They form even in the absence of transcriptional activity and its concomitant strand dissociation; hence, non-canonical structures may form spontaneously by competing with

the duplex (Shen *et al.*, 2021). (iv) G-quadruplexes and, possibly, *i*-motifs modulate transcription by acting as binding sites for transcription factors (Spiegel *et al.*, 2021; Zanin *et al.*, 2023; Esain-Garcia *et al.*, 2024). It follows that non-canonical four-stranded structures may form in the promoter, overcoming the constraints of Watson–Crick base pairing prior to the recruitment of RNA polymerase. (v) Transcription increases as the thermodynamic stability of a G-quadruplex in the promoter region increases, consistent with a shift in the duplex–tetraplex equilibrium toward the G-quadruplex conformation (Chen *et al.*, 2024).

Below, we expand on this line of reasoning and discuss each of the foregoing considerations in more detail. We first summarize our own biophysico-chemical results on duplex-tetraplex equilibria and then present an overview of a broader picture that emerges from molecular and cellular studies in other laboratories. Our focus is on the thermodynamics and, to a lesser extent, the kinetics of duplex-tetraplex interconversions within promoter DNA. Of particular interest is the transcriptional response to specific distributions of canonical and non-canonical DNA conformations in promoter regions of genes. While it is recognized that those effects are but one part of a multilayered regulatory process and operate in concert with other components of the transcriptional machinery, they are discussed here without explicit reference to the crucial role of intervening steps, which include other DNA regulatory elements, epigenetic modifications, chromatin accessibility, RNA polymerase, transcription factors, mediator proteins, and much else. An understanding of all steps is required if we eventually are to understand the relative place and importance of conformational heterogeneity of promoter sequences in the chain of events leading to transcription.

## Canonical and non-canonical conformations coexist in dynamic equilibrium, with fractional populations depending upon DNA sequence and environmental conditions

In the genome, the folding of a G-quadruplex or *i*-motif occurs in the presence of the complementary DNA strand. This proximity establishes a competition between the double-stranded and four-stranded states, resulting in a distribution of conformational states that may range from overwhelmingly duplex to overwhelmingly tetraplex. In other words, being rich in guanine and cytosine does not necessarily endow a particular genomic domain with the ability to break spontaneously from the constraints of Watson-Crick base pairing and form four-stranded structures.

Biophysical studies on duplex-tetraplex competition in G- and C-rich DNA molecules *in vitro* have shown that, when mixed together, complementary DNA strands bearing the human telomeric sequence adopt exclusively the duplex conformation; G-quadruplex or *i*-motif conformations are virtually nonexistent (Chalikian *et al.*, 2020). In contrast, G- and C-rich promoter sequences may adopt tetrahelical conformations that coexist in thermodynamic equilibrium with the duplex conformation (Chalikian *et al.*, 2020). The main challenge in such studies is to quantify the distribution of conformational states.

To address this problem, we have developed a CD spectroscopy-based procedure to determine the fractional populations of the duplex, G-quadruplex, *i*-motif, and coiled conformations in mixtures comprising equimolar amounts of G- and C-rich strands of DNA (Liu *et al.*, 2020, 2022). The procedure presupposes that the observed CD spectrum of such a mixture is the weighted sum of the "pure" spectra of the constituent conformations (Liu *et al.*, 2020). Each of the latter is generated and recorded independently, which

allows each observed spectrum to be unmixed in terms of the predetermined spectra of the constituent conformational states; that in turn allows one to obtain the corresponding weighting factors for the fractional contributions of those states to the total population of DNA (Liu *et al.*, 2020). The fractional values then can be analyzed in terms of the five-state models depicted in Scheme 1 to extract the thermodynamic parameters for the transition from each of the four fully or partially folded states to the unfolded, coiled state (Liu *et al.*, 2020).

We have investigated two structural arrangements: a bimolecular system in which the G- and C-rich strands are mixed in equimolar amounts (Scheme 1a), and a monomolecular system in which the two strands are joined by a covalent link (Scheme 1b) (Liu *et al.*, 2020, Liu 2022; Garabet *et al.*, 2025). In both systems, the conformational propensities of DNA have been studied as a function of temperature and the concentration of KCl at neutral and slightly acidic pH (Liu *et al.*, 2020, 2022; Garabet *et al.*, 2025). Weak acidity is conducive to the formation of an *i*-motif (Benabou *et al.*, 2014; Day *et al.*, 2014; Alba *et al.*, 2016; Tateishi-Karimata and Sugimoto, 2020).

In a bimolecular system, the formation of a tetraplex requires the dissociation and spatial separation of two complementary strands that otherwise would form a duplex. This contrasts with changes in the genome, where duplex-tetraplex transitions are pseudo-monomolecular in nature and are not accompanied by strand separation. From this perspective, monomolecular DNA constructs are better mimics of genomic DNA than are their bimolecular counterparts. Association of the two strands in a bimolecular system incurs a concentration-dependent translational entropic penalty. There is no such cost in monomolecular DNA, where the duplex-tetraplex equilibrium is shifted towards the duplex conformation relative to its iso-sequence bimolecular counterpart (Marky and Breslauer, 1987). This nuance is important in comparisons of conformational results obtained on mono- and bimolecular DNA constructs.

The bimolecular systems studied in our laboratory consist of complementary pairs of G- and C-rich DNA strands with sequences taken from the promoter regions of the c-MYC, VEGF, and Bcl-2 oncogenes (Liu *et al.*, 2020, 2022). Misregulated expression of those oncogenes is linked to the progression of a variety of cancers, including colon, ovarian, breast, prostate, pancreatic, and small-cell lung cancers, as well as osteosarcomas, leukemias, and lymphomas (Baretton *et al.*, 1996; Wierstra and Alves, 2008; Gonzalez and Hurley, 2010; Goel and Mercurio, 2013). The monomolecular system in our studies is a hairpin in which the complementary G- and C-rich strands of the stem are linked *via* a $dT_{11}$ loop and feature a sequence from the promoter region of the c-MYC oncogene (Garabet *et al.*, 2025).

In each system, the populations of the duplex, G-quadruplex, *i*-motif, and coil conformations engage in a complex exchange that is modulated by temperature, pH, and the concentration of potassium ions *via* changes in the differential free energies of the conformers (Liu *et al.*, 2020, 2022; Garabet *et al.*, 2025). Metal ions of an appropriate size, such as potassium, are an integral part of G-quadruplex structures; accordingly, an increase in the concentration of potassium ions causes an increase in the stability of a G-quadruplex (Lane *et al.*, 2008). It is noteworthy that the hairpin DNA adopts G-quadruplex-containing states only when $K^+$ and tetrabutylammonium ($TBA^+$) ions are present together in the buffer, both at pH 5.0 and at pH 7.0 (Garabet *et al.*, 2025). This observation is consistent with the selective binding of tetraalkylammonium ions to the parallel c-MYC G-quadruplex (Li *et al.*, 2024).

Temperature dependences of the fractional populations of the duplex, G-quadruplex, *i*-motif, and coiled states adopted by the double-stranded and hairpin constructs described above are shown at pH 5.0 and 7.0 in Figures 1 and 2, respectively. The curve for each state was computed according to the five-state model depicted in Scheme 1a (c-MYC-, VEGF-, and Bcl-2-based double-stranded DNA) or in Scheme 1b (c-MYC-based hairpin DNA). The required parametric values were those estimated as described above by deconvolution of the temperature-dependent CD spectra and subsequent analyses of the resulting fractional populations (Liu *et al.*, 2020, 2022; Garabet *et al.*, 2025).

At pH 5.0, which is the optimum pH for *i*-motif stability (Benabou *et al.*, 2014; Day *et al.*, 2014; Alba *et al.*, 2016; Kim and Chalikian, 2016), all four constructs were found to sample the full range of interconverting duplex, G-quadruplex, *i*-motif, and coiled conformations in proportions that depended upon the sequence and the effect of temperature and the concentration of potassium ions on each equilibrium (Figure 1) (Liu *et al.*, 2020, 2022; Garabet *et al.*, 2025). At physiological pH, the constructs adopted only the duplex and G-quadruplex conformations in proportions that

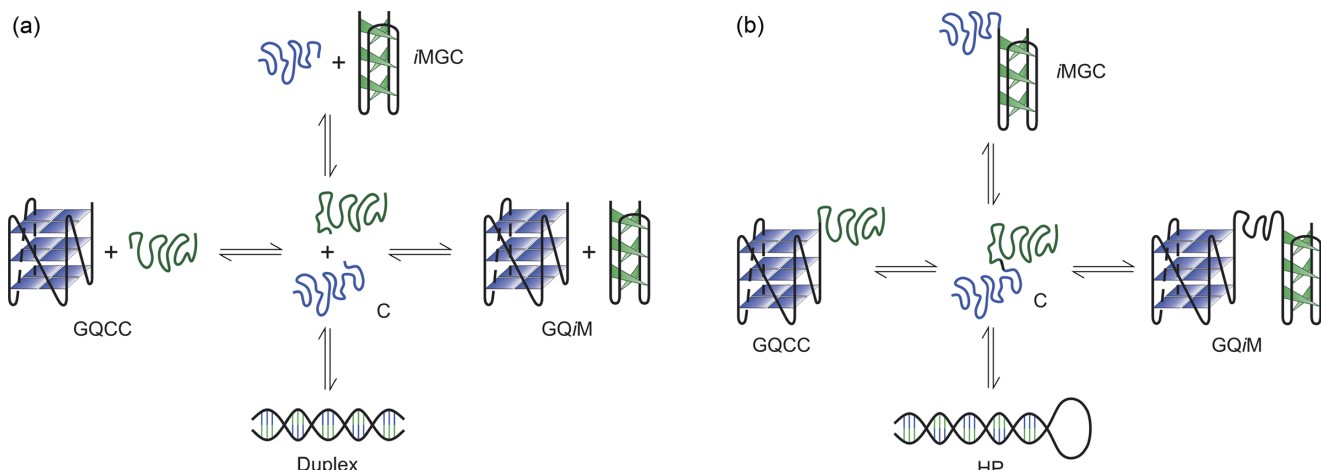

**Scheme 1.** (a) Equilibria linking the various conformational states adopted by an equimolar mixture of G-rich and C-rich strands of DNA (C, G-coil plus C-coil; D, duplex; GQCC, G-quadruplex plus C-coil; iMGC, *i*-motif plus G-coil; GQiM, G-quadruplex plus *i*-motif); (b) Equilibria linking the various conformational states adopted by the hairpin DNA (C, G-coil-plus-C-coil; HP, hairpin duplex; GQCC, G-quadruplex-plus-C-coil; iMGC, *i*-motif-plus-G-coil; GQiM, G-quadruplex-plus-*i*-motif).

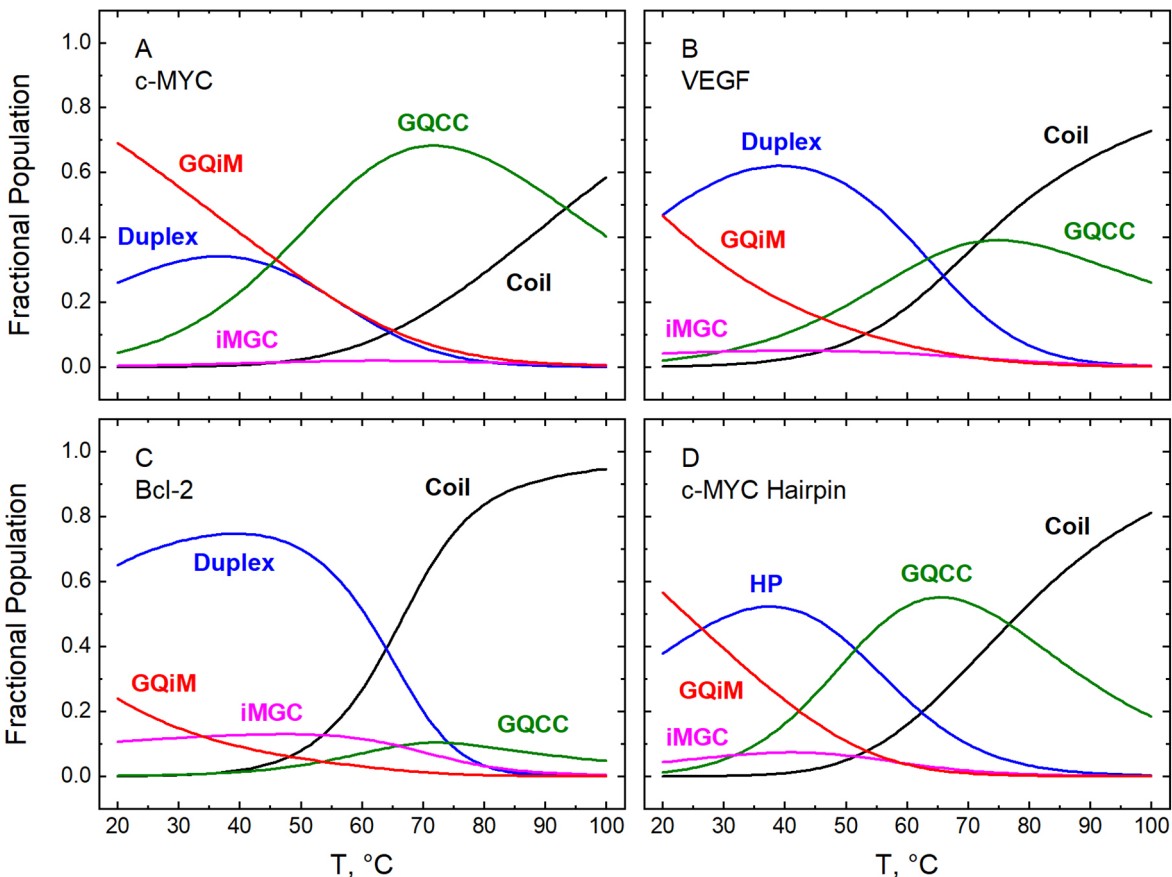

**Figure 1.** Fractional populations of the conformational states adopted at pH 5.0 by bimolecular constructs based on the promoters of three oncogenes (c-MYC, panel A; VEGF, panel B; and Bcl-2, panel C) and a monomolecular construct (hairpin) based on the c-MYC promoter (panel D). Values plotted on the ordinate were calculated according to Scheme 1a (bimolecular constructs) or Scheme 1b (hairpin) using thermodynamic parameters reported previously, as follows: panel A, Tables 1 and 2 in (Liu *et al.*, 2020); panels B and C, Tables 1–4 in (Liu *et al.*, 2022); panel D, Tables 1 and 2 in (Garabet *et al.*, 2025). Data were acquired in 50 mM KCl over the range of temperature shown on the abscissa.

similarly depended upon the sequence, the temperature, and the concentration of potassium (Figure 2) (Liu *et al.*, 2020, 2022; Garabet *et al.*, 2025). These observations are consistent with some reports that G- and C-rich DNA molecules can be distributed among different conformational states, in contrast to the notion that they exist overwhelmingly in a single conformation (Phan and Mergny, 2002; Chalikian *et al.*, 2020; Pandey *et al.*, 2023). Although we have not observed any significant presence of the *i*-motif conformation at pH 7.0, the situation may be different in the crowded environment of the cell. This possibility is supported by the stabilizing effect of molecular crowders on the *i*-motif, by reports that some DNA sequences can fold into an *i*-motif at neutral pH even in dilute (crowder-free) solutions, and by the visualization of stable *i*-motif structures in the cell (Nakano *et al.*, 2014; Wright *et al.*, 2017; Dzatko *et al.*, 2018; Zeraati *et al.*, 2018; King *et al.*, 2020; Takahashi and Sugimoto, 2020; Zanin *et al.*, 2023).

Our data and similar results from other laboratories offer persuasive evidence that, under appropriate conditions, complementary G- and C-rich DNA strands in bimolecular and monomolecular constructs form four-stranded G-quadruplex and *i*-motif structures that coexist in equilibrium with the duplex conformation (Chalikian *et al.*, 2020). The specific fractional ratios of duplex to tetraplex depend upon the DNA sequence and environmental conditions. Extrapolation of these *in vitro* facts to the cell suggests that four-stranded conformations may form spontaneously in the genome within the constraints of Watson-Crick base pairing and exist in

thermodynamic equilibrium with the duplex conformation. This idea forms the basis for our hypothesis, articulated below, that transcriptional regulation includes an important element of thermodynamic control.

## Statistical thermodynamic models describe observed conformational equilibria

Our data on the temperature-dependent conformational status of bimolecular and monomolecular DNA have been analyzed according to a statistical thermodynamic representation of the equilibria depicted in Schemes 1a and 1b (Liu *et al.*, 2020, 2022; Garabet *et al.*, 2025). Best fits of the model show consistent agreement with data obtained with different constructs under various conditions; that in turn is consistent with the underlying supposition that the exchange between double-helical and tetra-helical conformations adopted by complementary DNA strands originates in their differential thermodynamic stabilities (Liu *et al.*, 2020, 2022; Garabet *et al.*, 2025). The successful application of a model based on thermodynamic principles precludes the need for recourse to kinetics-based explanations; rather, the temperature-, pH-, and KCl-induced interconversions of the constructs in our studies can be described as thermodynamic in nature, involving thermodynamically stable states and not kinetically trapped, metastable intermediates. The importance of kinetic effects in the formation of tetraplex

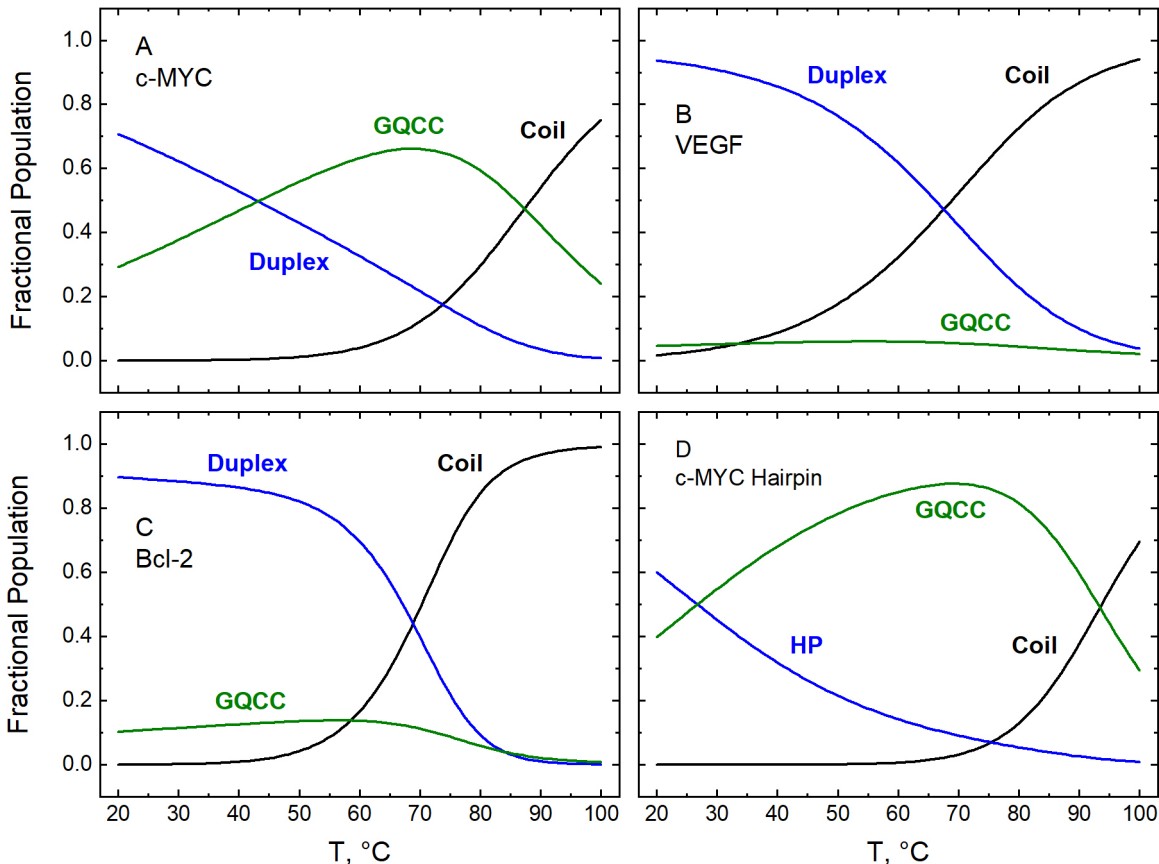

**Figure 2.** Fractional populations of the conformational states adopted at pH 7.0 by bimolecular constructs based on the promoters of three oncogenes (c-MYC, panel A; VEGF, panel B; and Bcl-2, panel C) and a monomolecular construct based on the c-MYC promoter (panel D). Values plotted on the ordinate were calculated according to Scheme 1a (bimolecular constructs) or Scheme 1b (hairpin) using thermodynamic parameters reported previously, as follows: panel A, Tables 1 and 2 in (Liu *et al.*, 2020); panels B and C, Tables 1–4 in (Liu *et al.*, 2022); panel D, Tables 1 and 3 in (Garabet *et al.*, 2025). Data were acquired in 50 mM KCl over the range of temperature shown on the abscissa. No line is shown for either of the *i*-motif-containing states (*i.e.*, *i*MGC and GQ*i*M), which are not populated at pH 7.0.

structures and the regulatory role of such effects in the genome is discussed below.

The observed equilibria between the double-helical and tetra-helical conformations of G- and C-rich DNA molecules *in vitro* (Chalikian *et al.*, 2020; Liu *et al.*, 2020, 2022; Garabet *et al.*, 2025) argue that, in the genome, non-canonical four-stranded conform-ations fold and unfold spontaneously in competition with the duplex. They are not restricted to preformed single-stranded DNA domains such as might originate in a prior genomic process (*e.g.*, within a transcription or replication bubble or an R-loop) (Crossley *et al.*, 2019; Chakraborty, 2020; Miglietta *et al.*, 2020; Petermann *et al.*, 2022; Wulfridge and Sarma, 2024). Narrowing this argument to a single DNA molecule in chromatin suggests that a G- and C-rich domain in the genome may fluctuate between a duplex on the one hand and a G-quadruplex or possibly an *i*-motif on the other in an equilibrium that depends upon the nucleotide sequence and intracellular conditions. Note that, in chromatin, the nucleosome protects Watson–Crick base pairing from disruption, which renders nucleosome-depleted G- and C-rich regions of DNA likely sites for the formation of G-quadruplex structures (Hansel-Hertsch *et al.*, 2016; Hansel-Hertsch *et al.*, 2018; Zhang *et al.*, 2023).

The notion of thermodynamically controlled duplex–tetraplex fluctuations within a single molecule *in vivo* is supported by an observed increase in the G-quadruplex population induced by G-quadruplex-binding drugs (Balasubramanian *et al.*, 2011; Husby *et al.*, 2013; Di Antonio *et al.*, 2020; Varshney *et al.*, 2020).

According to the principle of conformational selection, the binding of a G-quadruplex-selective ligand to a G-rich DNA domain implies the existence of some G-quadruplex in that domain in the absence of the ligand, although the tetraplex-to-duplex frac-tional ratio may be small (Vogt and Di Cera 2012, 2013; Vogt *et al.*, 2014; Chakraborty and Di Cera, 2017). In general, the dynamics of duplex–tetraplex transformations in DNA are expected to be affected by replication, transcription, and damage repair, all of which involve disruption of the duplex and are influenced by G-quadruplex-binding proteins such as helicases and transcription factors (Brazda *et al.*, 2014; Petermann *et al.*, 2022; Shu *et al.*, 2022; Zhang *et al.*, 2023).

The biological relevance of conformational fluctuations is well established in the case of proteins, with a high-energy and therefore sparingly populated conformation often being the functionally active one (Akasaka, 2006; Mittermaier and Kay, 2006; Sekhar and Kay, 2019). In a similar vein, hybridization–dehybridization dynamics have been reported for DNA duplexes (Ashwood and Tokmakoff, 2025), and the concept can be extended to duplex–tetraplex conformational dynamics in G- and C-rich domains of genomic DNA. That in turn has implications for the conform-ational control of genomic events, with transcription being just one example.

A word of caution is in order when thermodynamic insights gleaned from studies conducted on relatively short DNA constructs, either bimolecular or monomolecular, are applied to the

conformational dynamics of G- and C-rich sequences embedded in much longer, genomic DNA. The formation of a G-quadruplex or an *i*-motif in a long stretch of DNA may require separation of the strands in a region that is significantly longer than the sequence constituting the newly formed tetraplex. At one extreme, the number of base pairs that must be disrupted to enable a G-quadruplex to form may be on the order of the cooperative melting unit: that is, on the order of ~100 base pairs (Blake, 1987). The disparity between a large unfavorable change in free energy accompanying disruption of such a long duplex and a favorable change in free energy accompanying the formation of a much shorter G-quadruplex or *i*-motif may skew the duplex–tetraplex equilibrium in favor of the duplex.

## Conformational control of transcription

Under appropriate conditions, each promoter DNA in our investigation was found to establish an equilibrium in which the inter-converting duplex and G-quadruplex conformations are both present in appreciable amounts (Liu *et al.*, 2020, 2022; Garabet *et al.*, 2025). This suggests that at least some G- and C-rich promoter sequences sample canonical and non-canonical conformations spontaneously, with their fractional populations determined by the sequence. Based on these observations, we have put forward a hypothesis in which the transcription of a gene with a G- and C-rich promoter is regulated *via* the equilibrium between the duplex and G-quadruplex conformations, which is fine-tuned in a gene-specific manner to adjust the ratio of the two populations (Liu *et al.*, 2020, 2022; Garabet *et al.*, 2025). Transcription of a gene thereby is placed under thermodynamic control, with the G-quadruplex serving as an on- or off-switch for RNA polymerase activity. The hypothesis implies that a native ratio of conformations is unique to each gene. It follows that any deviation from that ratio, such as might arise from a mutation or a change in pH, potassium level, hydration, or other property of the system, may result in up- or down-regulation of the gene, potentially with pathological consequences.

G-quadruplexes may inhibit or enhance gene expression by sterically hindering the progression of RNA polymerase or by serving as recognition sites for G-quadruplex-binding proteins, such as transcription factors, that participate in transcription (Tateishi-Karimata and Sugimoto, 2020; Varshney *et al.*, 2020; Lago *et al.*, 2021; Robinson *et al.*, 2021; Shen *et al.*, 2021; Esain-Garcia *et al.*, 2024). The link between the formation of four-stranded DNA structures in G- and C-rich genomic domains and their involvement as enhancers or inhibitors of transcription is tightly controlled for each gene and cell cycle (King *et al.*, 2020). Fine-tuning of the G-quadruplex-to-duplex ratio in a promoter site is achieved by the combined effect of the natural conformational propensities of the site (the primary factor) and the timely intervention of G-quadruplex-binding proteins (secondary factors) (Mendoza *et al.*, 2016; Shu *et al.*, 2022). Hence, the physico-chemical exploration of the natural conformational propensities of genomic sequences is fundamental to an understanding of the conformational control of transcription and other genomic events.

The hypothesis articulated here provides a tool with which to explore the conformational control of genomic events such as transcription, and thereby to understand the mechanisms underlying a pathological under- or overexpression of a gene. For example, one or more mutations in a promoter sequence accompanied by disease-induced changes in cellular conditions, such as a decrease in pH or misregulation of the concentration of potassium

ions (Tateishi-Karimata *et al.*, 2018; Tateishi-Karimata and Sugimoto, 2021), may perturb the distribution of that sequence between its duplex and tetraplex conformations. Such a deviation from the norm may lead in turn to a change in oncogene expression.

## Results of cellular studies are consistent with the spontaneous formation of tetraplexes and thermodynamic control

Hundreds of thousands genomic sequences potentially can fold into a G-quadruplex (Chambers *et al.*, 2015). In contrast, much smaller numbers of folded G-quadruplexes—on the order of ~10,000—emerge from genome-wide mapping carried out by probe-based (Chip-Seq, CUT&Tag, Chem-map) and probe-independent (G4access) procedures (Galli *et al.*, 2024). One explanation for the vast difference between the number of G-quadruplex-forming motifs and the number of folded G-quadruplexes in the genome is that the G-quadruplex conformation is not accessible to a G-quadruplex motif within a nucleosome: G-quadruplexes form overwhelmingly in nucleosome-free regions of chromatin (Shen *et al.*, 2021). Another possibility emerges from our physico-chemical studies on oligomeric DNA (Liu *et al.*, 2020, 2022; Garabet *et al.*, 2025). In many G- and C-rich genomic domains, the equilibrium between the duplex and G-quadruplex conformations may be shifted markedly towards the duplex, with fractional populations of ~1% or less for the G-quadruplex.

Folded G-quadruplexes have been found predominantly in the regulatory domains of transcriptionally active genes, particularly in open chromatin regions (Hansel-Hertsch *et al.*, 2016; Hansel-Hertsch *et al.*, 2018; Lago *et al.*, 2021; Zanin *et al.*, 2023; Zhang *et al.*, 2023). Although less abundant than G-quadruplexes, folded *i*-motifs also tend to occur in those genomic regions and can overlap with R-loops, which underscores the interplay between the two tetraplex structures. This latter tendency turned up in a genome-wide mapping of *i*-motifs, where it was shown by means of the CUT&Tag procedure that G-quadruplexes and *i*-motifs may fold independently in different genomic loci or in the same sequence domain (Zanin *et al.*, 2023). Owing to the sensitivity limit of the CUT&Tag method, it is not possible to know if the G-quadruplex and *i*-motif form in parallel on complementary strands of the same genomic site in the same cell or in a mutually exclusive manner in different cells.

The formation of a G-quadruplex or an *i*-motif in a G-and C-rich genomic domain requires displacement of the complementary strand. That raises the question of whether a G-quadruplex in the genome forms spontaneously, in competition with the duplex (Scheme 2a), or consequentially, following separation of the duplex in a prior event such as the progression of RNA polymerase (Scheme 2b). In the latter case, the G-quadruplex would constitute a kinetically trapped, metastable state. Significantly, the importance of metastable states of DNA and the attendant interplay between long- and short-term kinetic intermediates in determining the global conformational distribution has been demonstrated recently by Breslauer and colleagues (Völker *et al.*, 2024).

Watson–Crick base pairing is a major obstacle to the formation of a G-quadruplex in the genome (Kim, 2019). Hence, the formation of a G-quadruplex in the genome typically has been thought to occur within the transcription bubble or R-loop that is formed transiently during strand separation and is coincident with RNA synthesis (Scheme 2b) (Kouzine *et al.*, 2004; Kim, 2019). In support of this mechanism, R-loop mapping along the genome has revealed that an R-loop at a single locus can extend over several hundreds

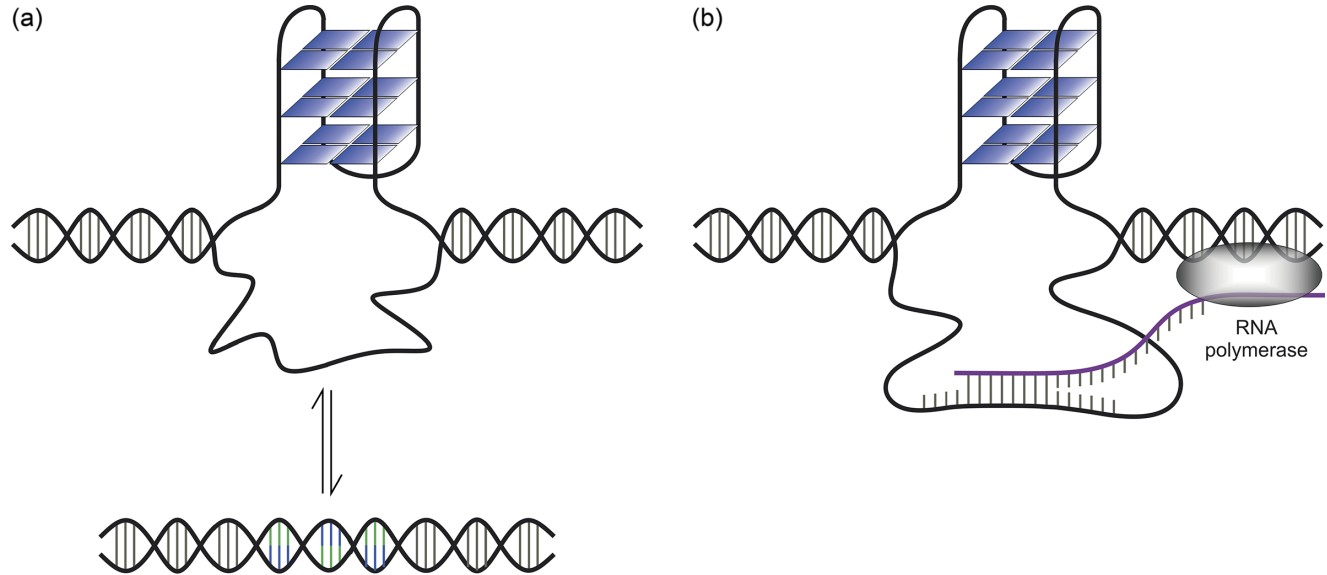

**Scheme 2.** (a) Spontaneous formation of a G-quadruplex in equilibrium with the duplex conformation; (b) Formation of a G-quadruplex within a single strand of DNA separated from its complementary strand in an R-loop.

base pairs (Chakraborty, 2020; Chedin and Benham, 2020). This greatly exceeds the 21–36 base pairs that is the mean length of a G-quadruplex in the genome, as identified by Chip-Seq, and the length of the most stable G-quadruplexes characterized *in vitro* (Lago *et al.*, 2021). In addition, R-loops and G-quadruplexes frequently are found together in genome-wide mapping of those structures (Lyu *et al.*, 2022). Further information on the structural interplay between G-quadruplexes and R-loops, *in vitro* and *in vivo*, and its functional consequences can be found in recent reviews (Miglietta *et al.*, 2020; Wulfridge and Sarma, 2024).

Kinetic effects notwithstanding, thermodynamic control of genetic processes implies that four-stranded secondary structures form not only consequentially within displaced single-stranded DNA but also spontaneously in competition with the duplex (Scheme 2a). Indirect evidence for duplex–tetraplex competition without prior separation of the duplex includes a recent observation that the inhibition of transcription does not result in the elimination of promoter G-quadruplexes in chromatin (Shen *et al.*, 2021). This finding suggests that strand separation within a transcription bubble or an R-loop is not a requirement for G-quadruplex folding. It similarly implies that the negative superhelicity that accompanies the progression of RNA polymerase also is not a requirement. The latter point is supported by results from one biophysical study in which supercoiling was found to have only a moderate effect on the formation of G-quadruplexes in a plasmid (Sekibo and Fox, 2017).

G-quadruplexes are not merely steric obstacles blocking the activity of RNA polymerase, a comparatively simple role that could be inferred from their exclusive localization in preformed, single-stranded stretches of DNA (Robinson *et al.*, 2021). In fact, they stimulate and enhance transcription by acting as sites for the recruitment of transcription factors to the promoter regions of actively transcribed genes (Robinson *et al.*, 2021; Spiegel *et al.*, 2021; Chen *et al.*, 2024; Esain-Garcia *et al.*, 2024). This is an important consideration, inasmuch as transcription factors typically associate with the promoter DNA prior to the recruitment of RNA polymerase and subsequent initiation of transcription and attendant strand separation (Alberts *et al.*, 2022). As alluded to above, the principle of conformational selection predicts that some G-quadruplexes always are present to some degree in the promoter

region and act as sites for the binding of transcription factors (Vogt and Di Cera, 2013; Vogt *et al.*, 2014; Chakraborty and Di Cera, 2017). Formation of a G-quadruplex, therefore, must be a spontaneous process that involves competing with the duplex, at least within some sequences, and it may or may not be linked thermodynamically to the binding of a transcription factor.

Further evidence that G-quadruplexes in promoters form by outcompeting the duplex is the observation that levels of transcription increase with an increase in the thermodynamic stability of the G-quadruplex in the promoter region (Chen *et al.*, 2024). Such an increase in stability implies an attendant shift in the duplex–tetraplex equilibrium towards the G-quadruplex conformation, in accord with the notion of thermodynamic control.

Evidence from a G4-Chip-seq/RNA-seq analysis of liposarcoma cells and keratinocytes suggests that a folded G-quadruplex, and not just a GC-rich sequence alone, is the binding site for transcription factors such as AP-1 and SP1 (Lago *et al.*, 2021). In that study, a comparison of data obtained from both types of cells revealed that the G-quadruplexes in the promoter of a gene are folded when the gene is actively expressed and mainly unfolded when it is downregulated (Lago *et al.*, 2021). The authors proposed that the transcription factors bind only to a folded G-quadruplex and that the folding or unfolding of a G-quadruplex within a promoter is a mechanism for controlling transcription in active genes (Lago *et al.*, 2021). Given that transcription factors associate with the promoter DNA prior to the recruitment of RNA polymerase, their suggested mechanism is consistent with the spontaneous formation of a G-quadruplex in competition with the duplex. This suggestion agrees with and complements the observation in human chronic myelogenous leukemia cells that the loss of promoter G-quadruplexes due to hypoxia-induced chromatin compaction is accompanied by the loss of RNA polymerase II binding to those same promoters (Shen *et al.*, 2021).

## Kinetic considerations and future developments

The folding of a G-quadruplex from its single-stranded conformer can be notoriously slow, which denotes folding intermediates separated by high-energy barriers (Gray and Chaires, 2008; Lane *et al.*,

2008; Gray *et al.*, 2014; Nicholson and Nesbitt, 2023; Lacen *et al.*, 2024). *In vitro* studies on model oligonucleotides suggest that G-quadruplex–duplex transitions also may be slow (Shirude and Balasubramanian, 2008; Mendoza *et al.*, 2015). These considerations draw attention to an important question concerning the interdependence of the thermodynamic contribution to transcriptional control and the kinetics of duplex–tetraplex interconversions in the genome. That question is centered on differences in the mechanisms and activation energies between the canonical and non-canonical states of DNA. Indeed, it has been suggested that, given the sluggish kinetics of interconversions between the thermodynamically stable and metastable conformers, metastable *i*-motif species may be more biologically relevant than their thermodynamically stable counterparts (Skolakova *et al.*, 2023).

Although duplex–tetraplex interconversions may be slow on the timescale of biological events, the equilibrium nevertheless can be treated as dynamic. Ratios of tetraplex to duplex *in vivo* can be increased by small ligands, G-quadruplex-binding proteins, and G-quadruplex-selective antibodies, as noted above, and that alone argues in favor of a dynamic equilibrium between the states. In one example, the stabilization of G-quadruplexes in a promoter by pyridostatin led either to up-regulation or to down-regulation of gene expression in a pattern that is consistent with a causal link between the amount of G-quadruplex and G-quadruplex-mediated activation or inhibition of transcription (Lam *et al.*, 2013).

While much evidence points to equilibrium-based control, one cannot exclude the possibility that kinetically stabilized G-quadruplexes occur *in vivo* and have biological significance. They may form within single-stranded stretches of DNA, such as those within a transcription bubble, and they may persist for a long time after the complementary strand becomes available for hybridization. The biological significance of kinetically stabilized states is widely acknowledged for proteins (Sanchez-Ruiz, 2010), and such states for G-quadruplex and *i*-motif structures may have a yet-unrecognized biological purpose.

A more complete understanding of the role played by thermodynamics in transcriptional regulation will require further thermodynamic and kinetic studies involving different DNA sequences, G-quadruplex topologies, molecular crowders, pH, and concentrations of potassium, all conducted *in vitro* and *in vivo* whenever possible.

Biophysical studies on many G- and C-rich promoter sequences are needed to provide information on their conformational propensities and the changes in those propensities caused by strategically introduced mutations or different environmental conditions. Currently, the most direct and quantitative way to explore the conformational preferences of specific genomic sequences is to study G- and C-rich oligonucleotide duplexes. Among such studies are those of the sort carried out in our laboratory, where the aim has been twofold: to quantify the conformational propensities of genomic sequences in terms of the fractional populations of the duplex, tetraplex, and coil states under different conditions, and to elucidate the balance of thermodynamic forces governing transitions from one state to another (Fan *et al.*, 2011; Kim and Chalikian, 2016; Liu *et al.*, 2018; Liu *et al.*, 2020; Chalikian and Macgregor, 2021; Liu *et al.*, 2021, 2022; Garabet *et al.*, 2025).

Biophysical studies need to be complemented by cellular studies that explore the relationships between subtle modifications of a promoter sequence, its conformational response, and the level of transcription of a reporter gene. Linking biophysical results obtained on oligonucleotides *in vitro* to conformational equilibria and associated transcriptional effects in the cell also requires studies

into the role of chromatin proteins, G-quadruplex-selective proteins, supercoiling, the crowded intra-cellular environment, and much else. Such factors exert a modifying influence on the natural conformational preferences of tetraplex-forming sequences, which can be viewed as a fundamental property of genomic DNA. Exploration of those preferences is therefore a necessary step in the quest to understand the conformational control of transcription and other genomic events. Without knowing the propensities inherent in regions of genomic DNA, the role of other factors cannot be understood.

## Outlook: a thermodynamic hypothesis

Insights from biophysical and cellular studies have been used in this report to advance a hypothesis in which transcription is subject to thermodynamic control. G- and C-rich domains in many promoters fold into G-quadruplexes and, in some cases, into *i*-motifs. In some sequences, four-stranded structures compete successfully with the duplex conformation, leading to the coexistence of both conformations in a thermodynamic equilibrium. In other sequences, four-stranded structures form only within regions of single-stranded DNA, such as a transcription or replication bubble, that come into existence when the duplex is disrupted by a prior genomic event. The location, mode of formation, and conformational distribution of specific tetraplexes is controlled by thermodynamic and kinetic factors that are tailored to an individual gene. Any deviation from the "normal" distribution of conformations owing to a point mutation or a change in cellular conditions may lead to under- or over-expression of the gene, with potentially pathological consequences. Thus, an understanding of the conformational propensities of G- and C-rich domains within the genome is a precondition to understanding the functional role of specific conformations as sites of biological action. This can lead in turn to means whereby the conformational preferences of tetraplex-forming sequences are modulated for therapeutic benefit.

**Open peer review.** To view the open peer review materials for this article, please visit http://doi.org/10.1017/qrd.2025.10013.

**Acknowledgements.** We thank Professors Robert B. Macgregor, Jr., Jens Völker, Arno Siraki, and Jeffrey Henderson for many fruitful discussions.

**Author contribution.** J.W.W. and T.V.C. conceived and wrote the paper.

**Financial support.** This work was supported by Leslie Dan Faculty of Pharmacy of the University of Toronto.

**Competing interests.** The authors declare none.

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
