## [Reviewer Report]

The subject of this review is interesting and much of the content is fine. However, the authors need to ask “Who is my audience?” and “What is my storyline?” At the moment the manuscript cannot be read without either detailed knowledge of the relevant literature especially the authors' own work or a lot of parallel reading - which rather defeats the purpose of this Discovery Review type of paper. I have a few specific issues but basically the authors need to think about the reader. They need some good figures to illustrate what they are talking about and some more introductory material.

1. References for "The canonical B-DNA duplex is not the only conformation available to genomic DNA” are all 2017 ff – I remember reading about this long before that, though I can’t remember where.

2. c-MYC – perhaps include what its role is and why this is important. It rathe jumps out of nowhere.

3. I am really struggling to understand what the authors mean by Thermodynamic – to me it is obvious that DNA is a dynamic CONTINUUM of structures (not just 2-state) that are remarkably close in energy and that the environment can therefor effect the equilibrium. I assume this is what also underpins their thinking. I think the review would benefit from a short discussion of relevant DNA structures and their energy differences say in vacuo, in solvent, with and without ions, in presence and absence of proteins. Much more consideration needs to be given to environment when the review discusses adoption of structures - how much on an energetic edge is the system. What is the role of entropy? If the paper is about thermodynamic control what they mean by it needs to be clear and DATA need to presented to address the hypothesis.

4. The assumptions of 2-state for the spectroscopy analysis should be explored further.

5. Where are the data???

TO EDITOR: I really object to my review being published. It inhibits me helping the authors to improve their work.

---

## [Reviewer Report]

DNA structure is not static; it’s a dynamic, responsive scaffold that allows genetic information to be read. Ultimately, the structural dynamics determine which genes are turned on or off, when, and how strongly. This structural dynamic is intricately regulated by the physical and chemical structure of DNA, especially how it’s packaged and modified within the cell. It involves several levels of regulation. There is a chromatin architecture that is controlled by interactions with the histone proteins. It can be either loosely packed, forming euchromatin, where genes are accessible and can be transcribed. Or it can be tightly packed, forming heterochromatin, where genes are silenced because transcription machinery can’t reach them. Epigenetic modifications, such as DNA methylation and post-translational histone modifications, can further modulate the transition between euchromatin and heterochromatin. DNA regulatory elements, including promoters, enhancers, silencers, and insulators, interact with transcription factors through specific DNA sequences. The location of these sequences and accessibility depend on DNA folding and looping, which in turn is influenced by chromatin structure.

The mechanism discussed in this paper differs from these well-known ways of regulating gene expression. It suggests that the actual structure of DNA changes in response to external cues such as temperature, pH, or concentration of specific ions. The authors propose a regulatory role for conformational switches between canonical duplex DNA and non-canonical structures (G-quadruplexes and i-motifs) that can be formed by G- and C-rich genomic regions. Experimental data indicate that pH, ion concentration, and temperature alter the populations of different local structural motifs, and a strict thermodynamic formalism can rationalize these populations. They show that even point mutations can significantly change the distribution of states. Furthermore, they provide evidence from genome-wide mapping that nucleosome-free regions of actively transcribed genes contain sequences that form G-quadruplexes.

Overall, the hypothesis is well substantiated and well-received. The authors emphasize the need for further studies to understand the thermodynamic and kinetic factors governing DNA conformational dynamics and their implications for transcriptional control and therapeutic applications. Within these lines, my only minor quibble is a request for the author to provide a more unambiguous indication of what type of information, in their opinion, is missing to further substantiate and/or refute the hypothesis they put forward.

---

## [Reviewer Report]

The manuscript is much easier to read now and I have recommended acceptance. However, the manuscript subject is a very visual and structural topic and I am surprised that there are 10 pages of text before any figure to illustrate the text is included. What is G and C and why are the special? What is an I-Motif? What is a quadruplex? The authors are experts and don’t need visual help. If their aim is to teach people who do not work with DNA structures, a figure or two early on would really help the kind of reader who turns to a review.